# Comparative Study of Aryl *O*-, *C*-, and *S*-Mannopyranosides as Potential Adhesion Inhibitors toward Uropathogenic *E. coli* FimH

**DOI:** 10.3390/molecules24193566

**Published:** 2019-10-02

**Authors:** Leila Mousavifar, Gérard Vergoten, Guillaume Charron, René Roy

**Affiliations:** 1Department of Chemistry, Université du Québec à Montréal, P.O. Box 8888, Succ. Centre-Ville, Montréal, Québec H3C 3P8, Canada; charronguy@hotmail.com; 2Glycovax Pharma Inc., 424 Guy, Suite 202, Montreal, Quebec H3J 1S6, Canada; 3Unité de Glycobiologie Structurale et Fonctionnelle (UGSF), UMR8576 du CNRS, Université de Lille, F-59000 Lille, France; Gerard.vergoten@univ-lille.fr; 4INRS-Institut Armand-Frappier, Université du Québec, 531 boul. des Prairies, Laval, Québec H7V 1B7, Canada

**Keywords:** carbohydrate, D-mannosides, uropathogenic infections, *E. coli*, lectin, FimH, Heck reaction, metathesis, SPR, X-ray

## Abstract

A set of three mannopyranoside possessing identical 1,1′-biphenyl glycosidic pharmacophore but different aglyconic atoms were synthesized using either a palladium-catalyzed Heck cross coupling reaction or a metathesis reaction between their corresponding allylic glycoside derivatives. Their X-ray structures, together with their calculated 3D structures, showed strong indicators to explain the observed relative binding abilities against *E. coli* FimH as measured by a improved surface plasmon resonance (SPR) method. Amongst the *O*-, *C*-, and *S*-linked analogs, the *C*-linked analog showed the best ability to become a lead candidate as antagonist against uropathogenic *E. coli* with a Kd of 11.45 nM.

## 1. Introduction

Multidrug resistant bacteria constitutes a serious public health burden with approximately 700,000 people dying each year from infections caused by antibiotic resistant organisms. This number is expected to reach 10 million people annually by 2050 with related cost ranging $100 trillion US dollars [1]. Although medicinal chemistry has allowed the relatively fast discoveries of diverse families of potent antibiotics working under a wide range of bactericidal or bacteriostatic mechanisms [2], bacteria have similarly developed a variety of ingenious defense mechanisms [3]. Therefore, there is an urgent need to fight bacterial infections by mechanisms that alleviate multidrug resistances. One of these approaches, showing increasing successes, is directed at preventing the early step of bacterial adhesion and colonization, and biofilm formation pathways [4]. Since several infection mechanisms are initiated by molecular recognition between the host cell surface receptors and virulence factors on the bacterial surfaces [5], such strategies are likely to be unavoidable without impeding the bacteria’s own capacity to recognize and adhere to the targeted tissues. Urinary tract infections (UTIs) represent one of the most common bacterial infections, affecting one out of two women during their lifetimes. UTIs are serious health problems as they reach close to $2.5 billion in annual healthcare costs in the US alone [6] which are caused by Gram-negative uropathogenic *Escherichia coli* (UPEC) strains [7].

UPEC are the major cause of urinary tract infections and intestinal bowel diseases (IBD) including Crohn’s disease (CD) [1,6]. These strains enter urinary tracts to form colonies. The adhesions of various pathogenic *E. coli* strains to host cells are primarily mediated through carbohydrate-protein interactions involving bacterial fimbriae which can recognize specific glycoconjugate receptors on host cells. Of particular interests are the FimH and the PapG fimbriae that bind to mannosylated glycoproteins or glycolipids, respectively [8].

The FimH bacterial lectins adhere to the highly mannosylated glycoprotein uroplakin 1a [9,10] on the bladder cells and on glycoprotein 2 (GP2) exclusively expressed in the small intestine [7,11]. Once the infection process is initiated, it is usually followed by thick biofilm formation [12,13]. Biofilms have a pronounced impact in clinical settings because they render bacterial infections resilient to antibiotics [14]. Given that bacterial adhesion to host cells is the preliminary step toward the release of toxic proteins, the design of potent *E. coli* FimH antagonists has been the target of several efforts [15,16]. Several glycomimetic analogs of the naturally occurring complex oligomannosides have been synthesized [8,10] that provided potent α-D-mannopyranoside-based antiadhesins. Since small molecule antagonists represent the foremost choice of the pharmaceutical industry, these glycomimetics constitute most promising candidates for the replacement of the natural lead oligomannosides onto which *E. coli* adhere to [10,17].

To circumscribe the effect of anomeric linkages upon the design of uropathogenic *E. coli* FimH antagonists based on mannopyranosides, we compare herein three candidates possessing aromatic aglycones capable to fit within the well-established hydrophobic tyrosine gate present in the mannopyranoside binding pocket. Hence, the tridimensional structures, relative binding affinities of *O-, C-,* and *S*-linked mannopyranosides harboring a common 1,1′-biphenyl pharmacophore are described.

## 2. Results and Discussion

### 2.1. Glycomimetic Synthesis

Since *O*-substituted mannopyranosides are susceptible to hydrolysis by mannosidases and low pHs of the gut, there is a need to identify alternative candidates suitable under in vivo conditions. *N*- [13], *C*- [10,18,19], or *S*-linked [20] glycomimetics have the potential to possess the required stabilities and appropriate pharmacokinetic and pharmacodynamic parameters.

Analogues of *O*- and *C*-linked α-D-mannopyranosides with hydrophobic and aryl substituents have already been identified as potent *E. coli* FimH antagonists having low nanomolar Kds or IC_50_s [5,10,18,21,22]. *Para*-substituted biphenyl derivatives were shown to be particularly appealing owing to their numerous favorable binding interactions within the identified tyrosine gate formed between Tyr48 and Tyr137 [18,21,23,24,25]. In turn, some derivatives were shown to bind better when the Tyr-gate is either open, half-open, or close, further pointing toward rationally design glycol therapeutics [20,22,26,27,28]. In fact, using the centroid positions of each of the Tyr48 and Tyr137 phenyl groups in crystal data sets, we calculated that the Tyr-gates are 8.77 Å, 7.79 Å, and 6.99 Å apart in the open (PDB 4AV0) [29], half-open (PDB 4BUQ) [29], and close states (PDB 4AUY) [30], respectively. Since this family of α-D-mannopyranoside derivatives showed high promises as antagonists, we decided to further evaluate their comparative 3-D structures, relative binding capacities, and biophysical characteristics, by exploring the roles played by substituting the anomeric linkages with heteroatoms, while keeping constant the *para*-biphenyl moieties. Toward this goal, we first accessed a series of α-D-mannopyranosides **1–6** (Figure 1) known to serve as comparative standards. Compounds **1** (MeαMan) and **2** (PNPαMan) are commercial products, while **3** (HeptylαMan), the golden standard with a reported Kd of 5 nM (SPR), together with compounds **4** (see Appendix A) were prepared by literature procedures. The 1, 1′-biphenyl derivatives with an anomeric *O*-linkage (**5**) and the corresponding *C*-linked analog (**6**), albeit with one atom shorter, were also prepared by published procedures [18]. Note that we successfully crystallized compounds **5** and **6** for the first time (see discussion below).

We describe herein the synthesis of the related *S*-linked α-D-mannopyranoside (**12**) (Scheme 1). Similarly to its *O*- and *C*-linked congeners, we chose the *S*-allyl α-D-mannopyranoside **9** as key starting materials. However, rather than using a palladium catalyzed Heck reaction and a series of aryl iodides for the cross-coupling as before [18,21], we decided to extend its alkenyl functionality by a metathesis reaction using 4-vinyl-1,1′-biphenyl (**10**) and Grubbs 2^nd^ generation catalyst. This alternative strategy was considered equally appealing owing to its potential to access a diversified library of analogs. In this way, we can demonstrate the versatility of these two complementary chemical strategies leading to these important families of *O*-, *C*-, and *S*-linked derivatives.

Eventhough the necessary intermediate **9** was claimed to be prepared in a single step from D-mannose [31], the procedure failed in our hands to provide reproducible and high yields. Hence, we relied on the more classical Ferrier’s conditions [32,33,34] from peracetylated D-mannose **8** (Scheme 1) which, when treated with prop-2-ene-1-thiol (allyl mercaptan) and BF_3_-Et_2_O in DCM, provided the α-anomer **9** in 53% yield after purification. All the physical data agreed with the published one [34], thus providing unambiguous evidence for an α-isomer. When **9** was further treated with 4-vinyl-1,1′-biphenyl (**10**) and Grubbs 2nd generation catalyst under refluxing DCM [24,35,36], the desired *trans* analog **11** was obtained in 56% yield after silica gel purification. Classical Zemplén deprotection (NaOMe, MeOH) afforded **12** quantitatively.

### 2.2. Structural and Conformational Analyses

Having secured entry to our three heteroatom-linked α-D-mannopyranosides **5**, **6**, and **12**, we evaluated their detailed 3D structures as it was anticipated that the relative orientation of their *p*-biphenyl pharmacophores with the *E. coli* FimH binding site would be determinant for their relative binding abilities. This is particularly true given the difference in the width of the Tyr gate in the open, half-open, and close states, as mentioned above and that they may individually bind to one form or another unpredictably. Fortuitously, we accessed to two crystalline structures, namely, those of the *O*- and *C*-linked analogues **5** and **6**, respectively. In addition, we previously resolved fourteen X-ray co-crystal data of several *O*-linked analogues bound to the *E. coli* FimH, thus giving us access to several docking opportunities [10,18,21,29,30]. One of them in particular (**7**) is of interest (PDB 4AV5) [29] as it also contains a 1,1′-biphenyl ring, albeit with a more rigid alkynyl linkage (see Appendix A for superimposed figure containing **5** and **7**). Interestingly, while **7** showed a Kd of 405 nM**, 5** was 135 fold better with a Kd of 3 nM (SPR) [18], again demonstrating the importance of pharmacophore orientation within the tyrosine gate. Appendix A clearly shows compound **5** forming a closer contact with Tyr137 in the open form. It’s docking within the FimH active site was done by superimposition of the mannoside residues with that of known **7**.

The crystalline structure of **5** alone is shown in Figure 2. Also of interest is the fact that the two biphenyl rings are not tilted with respect to one another. This is in strike contrast with the situation observed from other *p*-biphenyl candidates including our own above (**7**, PDB 4Av5) [28,29,37]. In addition; it is also noteworthy that the conformation around the C5-C6 bond in the mannose ring is in the *gg* orientation, as it is the case of most crystal structures of mannosides within the binding pocket of FimH.

The *C*-linked analogue **6**, having a C-O-C bond shorter, deserves some specific comments on its own because of the controversial arguments about the preferred conformational states of this family of glycomimetics. Indeed, several literature cases argued for the fact that *C-*linked α-D-mannopyranosides do not exist preferentially in the “usual” ^4^C_1_ chair conformations, at least for simple aglycones [10,38,39,40,41]. As already shown by us with *O*-linked allyl α-D-mannopyranoside [10], the ^4^C_1_ conformation was the most stable form with the inverted ^1^C_4_ chair being next in line with a difference of 8.10 kcal/mole difference. To see if this observation still holds for compound **6**, we undertook a conformational search using the Monte Carlo method [42]. The retained structure was then optimized using molecular mechanics [43,44,45] and the density functional theory (DFT) with the hybrid functional B3LYP, base 6-31G* (see Appendix A). The result clearly showed, once again that this additional *C*-linked α-D-mannopyranoside **6** also predominates in the desired ^4^C_1_ conformation.

Fortuitously again, we were able to crystallize compound **6** alone as they appear in Figure 3 as **6A** and **6B**. However, the situation was complicated by the fact that the crystal lattice showed two sugar residues stacking to one another (Figure 3, **6A** and **6B**). Apparently, both biphenyl moieties were forming strong CH-π interactions [10,46,47], and likely reinforced in that state since the C3-OH group (**6A**) was hydrogen bonded to the C6-O (**6B**, 3.914 Å). In addition, one sugar (**6A**) had the C5-C6 bond in the *tg* conformation while **6B** was in the *gg* state. Moreover, both of the *p*-substituted biphenyl rings were flat, as opposed to most observed cases discussed above [23,24] when the sugars are bound to the FimH. This is also seen in the calculated conformer ^4^C_1_ (Figure 3, **6C**) where the two rings are tilted by –56.0° in close agreement to those observed for PDB 3MCY [23] (–36.21°) and PDB 5F2F [23] (–37.47°) [24]. Additionally, the C5-C6 bond was in the *tg* conformation.

Clearly, for the situation seen when **6** crystallized alone, the self-assembly into a dimeric form becomes a dominant factor which could not be seen in either the calculated values nor with the FimH binding site. Other significant differences were observed between the three forms denoted **6A**, **6B**, and **6C** for the crystalline states and calculated one, respectively: ΦA = –35.84°, ΦB = –54.84°, ΦC = +54.48°; ΨA = –54.50°; ΨB = +65.63°, ΨC = –91.69, wherein the Φ torsion angle is define between H1-C1-C1′-C2′ and the Ψ torsion angle is define as C1-C1′-C2′-C2′H. 

The fact that FimH is known to adopt three different conformations at the tyrosine gate (Tyr48-Tyr137; *open, close, half open*) depending to which ligands it is bound to, constitutes an additional example of molecular adaptation that has been previously observed with *C*-linked oligomannoside analogues [47,48], a process of conformational selection suggesting that the FimH likely binds these glycomimetics similarly to the way that it binds the natural analogues [24,49].

Contrary to the above situation for the *O*- and *C*-linked analogues **5** and **6**, we could not obtain a crystalline structure for the *S*-linked analogue **12**. However, it is well known that thioglycosides also obey to the *endo*- as well as the *exo*-anomeric effects [50]. Therefore; we undertook to perform the above Monte Carlo and DFT molecular dynamic calculation on **12** to glance at its most likely 3D structure. As anticipated, the C1-S and the S-C1′ bonds length were 1.862 and 1.853 Å, respectively, longer than the equivalent C1-O and O-C1′ bonds lengths in **5** (1.399 Å, 1.496 Å). Moreover, the Φ was calculated to be –45.93° in good agreement with the *exo*-anomeric effect for **5** (Φ = –57.69°) (Figure 4).

### 2.3. Relative FimH Binding Affinity

It has been previously observed that the analogous *O-, C-* and S-linked heptyl α−D-mannopyranosides had IC_50_s in the low nanomolar range when measured by surface plasmon resonance (SPR) against the *E. coli* FimH [20]. However, the order was *O*- (IC_50_ = 160 nM), *C-* (IC_50_ = 424 nM), and the *S*-linked (IC_50_ = 547 nM), thus showing that the *O-*linked analogue was still unsurpassed and that the nature of the aglyconic heteroatoms had a critical effect on the binding affinities.

To evaluate the relative binding affinities of our own series of compounds **5**, **6**, and **12**, we undertook surface plasmon resonance (SPR) measurements using an improved competition method. To this end, we first immobilized 6-aminohexyl α-D-mannopyranoside (**4**) on the CM5 gold chip using the conventional carbodiimide method to which we allowed FimH to bind with and without our competitive inhibitors (see sensorgram of **4** in the Appendix A). The equilibrium dissociation constant (Kd) of the interaction was then normalized, the data sets are illustrated in Figure 5 and values given in Table 1 (see experimental details). This modification from the previous protocol [18] using 8-aminooctyl α-D-mannoside was found much more practical and reliable given its very high affinity to FimH which made the gold chip regeneration from sample to sample too complicated (very slow k_off_).

The binding data of methyl- (**MeαMan, 1**), *p*-nitrophenyl α-D-mannopyranoside (***P*N*P*αMan, 2**), and heptyl- (**heptylαMan, 3**), were also included for comparison purpose. Their relative binding affinities are in the same order as those usually measured by other SPR methods [46], thus justifying the validity of our new procedure. As can be seen, the *C-*linked analog **6** was a better ligand (Kd = 11.45 nM) than the corresponding *O*-linked (**5**) (Kd = 19.90) and the S-linked (**12**) (Kd = 94.4 nM). Given the subtle changes in the 3D structures between **5** and **12**, the difference in their respective Kds is somewhat surprising (about 5 fold). However, the situation is similar for the related series of *O-* and *S*-linked heptyl α-D-mannopyranosides (see above, 3.4 fold). The major difference here is that the *C-*linked analog **6** was better than the other two and close to some of the best known *E. coli* FimH antagonists [10,18]. This could be attributed to a better π-π stacking of the biphenyl substituent of **6** within the tyrosine gate.

## 3. Materials and Methods

Reactions were carried out under Nitrogen using commercially available ACS grade solvents which were stored over 4 Ǻ molecular sieves. Solutions in organic solvents were dried over anhydrous Na_2_SO_4_, filtered, and concentrated under reduced pressure. Reagents were obtained from Sigma Aldrich (Toronto, ON, Canada). The FimH residues 1-158 were isolated from the *E. coli* strain UTI89. The source was obtained from a synthetic gene expressed into pET24a [51,52,53]. Methyl α-D-mannopyranoside (**Me Man, 1**) and p*ara*-nitrophenyl α-D-mannopyranoside (**PNP Man, 2**) are commercially available from Alfa Aesar (Tewksbury, MA, USA) and Sigma Aldrich, respectively. Heptyl α-D-mannopyranoside (**Heptyl Man, 3**) was prepared according to published procedure [18,53]. Melting points were measured on a Fisher Jones apparatus and are uncorrected. Reactions were monitored by thin-layer chromatography using silica gel 60 F254 coated plates (E. Merck). NMR spectra (Merck, Darmstadt, Germany). Were recorded on Varian Inova AS600 and Bruker Avance III HD 600 MHz spectrometer. Proton and carbon chemical shifts (δ) are reported in ppm relative to the chemical shift of residual CHCl_3_, which was set at 7.28 ppm (^1^H) and 77.16 ppm (^13^C{^1^H}). Coupling constants (*J*) are reported in Hertz (Hz) and the following abbreviations are used for peak multiplicities: Singlet (s), doublet (d), doublet of doublets (dd), doublet of doublet with equal coupling constants (t_ap_), triplet (t), multiplet (m). Assignments were made using COSY (Correlated SpectroscopY) and HSQC (Heteronuclear Single Quantum Coherence) experiments. High-resolution mass spectra (HRMS) were measured with a LC-MS-TOF (Liquid Chromatography Mass Spectrometry Time of Flight) instrument (Agilent Technologies Santa Clara, CA, USA). In positive electrospray mode by the analytical platform of UQAM. SPR were performed with a Biacore T200 on a CM5 sensor chip (GE Healthcare Life Sciences, Mississauga, ON, Canada).

### SPR Solution Affinity of Ligands for the E. coli FimH Adhesin

First, the equilibrium dissociation constant (K_D_) of FimH against immobilized 6-aminohexyl α-D-mannopyranoside (**4**) was determined. The ligand 6-aminohexyl α-D-mannopyranoside (10 mM in H_2_O) was immobilized via standard amine carbodiimide coupling chemistry (and blocking with ethanolamine as per the manufacturer’s procedure) to obtain 364 response units (RU) (see sensorgram in Appendix A). The reference channel was treated similarly without ligand. A concentration series of the analyte FimH (1024 to 8 nM in running buffer, 2-fold dilutions) was then injected over the surface (700 s association time; 1200 s dissociation time; 30 μL/min flow rate; 25 °C temperature; running buffer: PBS pH 7.45, 3 mM EDTA supplemented with 0.005% Tween-20; regeneration: 30 s of 10 mM Glycine pH 1.5) and the response was double-referenced by subtracting the reference channel and the response with no FimH (running buffer only). Since equilibrium could not be reached during injection, K_D_ was obtained from the extracted kinetic rate constants (K_D_ = k_off_ / k_on_) from a non-linear globally fitted 1:1 Langmuir binding isotherm containing a mass transfer constant (t_c_) and locally fitted refractive index constant (RI) (Biacore T200 evaluation software v1.0, GE Healthcare Life Sciences) of the obtained sensorgrams.

Second, the equilibrium dissociation constants (K_D_) of FimH against the compounds of interest were determined by measuring the amount of free FimH in a series of concentrations of the compounds of interest at a constant concentration of FimH (close to the Kd of FimH: 6-aminohexyl α-D-mannopyranoside) after equilibrium in solution. A concentration series of the ligand of interest (65536 to 0.5 nM in running buffer, 2-fold dilutions) was equilibrated overnight with analyte FimH (40 nM) and then injected over the previous ligand 6-aminohexyl α-D-mannopyranoside-immobilized surface (300 s association time; 300 s dissociation time; 30 μL/min flow rate; 25 °C temperature; running buffer: 5% DMSO, PBS pH 7.45, 3 mM EDTA supplemented with 0.05% Tween-20; regeneration: 30 s of 10 mM Glycine pH 1.5) and the response was single-referenced by subtracting the reference channel and solvent corrected with a DMSO calibration curve. Since equilibrium could not be reached during injection, the responses (free [FimH], Y-axis) at the end of the injections were normalized with the response from FimH only ([compound of interest] = 0) and then plotted over the compounds of interest concentrations (X-axis). The obtained curves were then non-linearly fitted according to the following equation [53] using GraphPad Prism (v6.07, GraphPad Software, San Diego, CA, USA) to extract R_max_ and K_S_:$$Y=\left(\frac{\left[FimH\right]-A}{K_{C}+\left[FimH\right]-A}\right)R_{max}$$
$$A=\frac{\left[FimH\right]+X+K_{S}-\sqrt{{\left(\left[FimH\right]+X+K_{S}\right)}^{2}-4\left[FimH\right]X}}{2}$$
where: Y is the concentration of free FimH (RU); R_max_ is the binding capacity of the SPR surface at saturation of all binding sites; K_C_ and K_S_ are the equilibrium dissociation constants (K_D_) for binding of FimH to immobilized 6-aminohexyl α-D-mannopyranoside ligand on the sensor surface and with the compound of interest in solution, respectively; [FimH] is the constant concentration of FimH (40 nM); X is the concentration of the compound of interest (nM). 

## 4. Conclusions

The work presented herein which describes the syntheses, tridimensional structures, binding parameters using a modified SPR procedure of a series of *O*-, *C*-, and *S*-linked mannopyranoside derivatives incorporating the identical 1,1′-biphenyl pharmacophore but diverse aglyconic atoms illustrates the necessity to explore this aspect when exploring novel *E. coli* FimH antagonists. The fact that our series of biphenyl analogues differs from those previously obtained from the corresponding heptyl mannopyranosides, clearly justify the deepen analysis undertaken in this report. Clearly, this study also confirms the fact that our *C*-linked α-D-mannopyranoside unambiguously exists in its preferred ^4^C_1_ chair conformation as opposed to several claims in the literature. It also further confirms that *S*-glycoside to possess an *exo*-anomeric effect. Even though, none of the compounds exhibited higher affinities than the golden *O*-linked heptyl α-D-mannopyranoside standard, this work expanded our knowledge on the *E. coli* FimH tyrosine gate which showed that further binding contacts are worth exploring. This fact on its own strongly argues for revisiting some of the recent best antagonists identified thus far and that they can be further optimized. Further studies in vitro and PK/PD evaluation will warrant deepen our approach toward truly therapeutic candidates. 

*Allyl 2,3,4,6-tetra-O-acetyl-1-thio-α-D-mannopyranoside (***9***).* To a solution of peracetylated mannose (**8**), (405 mg, 1.12 mmol) in 2 mL of dried methylene chloride was added, under N_2_, allyl mercaptan (374 µL, 4 equiv.). The mixture was cooled at 0 °C and BF_3_Et_2_O (1.4 mL, 10 equiv.) was added. The reaction mixture was stirred at room temperature for 20 h. The reaction was quenched with 2 mL of water under stirring, then diluted with methylene chloride and was washed with 30 mL of 5% aqueous solution of NaHCO_3_. The organic layer then dried over Na_2_SO_4_, and concentrated under reduced pressure. The residue was purified by silica gel column chromatography using Hexane/ EtOAc (7:3) to afford compound **9** as a colorless oil with *Rf* = 0.35 (Hexane/ EtOAc (7:3)), (240 mg, 0.593 mmol, 53%). (c = 0.3, CHCl_3_); [α]^20^_D_ = 66.1, m.p. 66.5–69 °C. ^1^H NMR (300 MHz, CDCl_3_):, 5.84–5.76 (m, 1 H, CH=CH_2_), 5.36–5.30 (m, 3H, H-4,H-3,H-2), 5.29–5.19 (m, 3H, H-1, CH=CH_2_), 4.42–4.35, (m, 2H, H-5,H-6a,), 4.11 (dd, 1H, *J*_6a, 6b_ = 14.8 Hz, *J*_5, 6a_ = 8.4 Hz, H-6b), 3.29–3.15 (m, 2H, H1′a,1′b), 2.18, 2.1, 2.06, 2.00 (4s, 12H, 4×COCH_3_),. ^13^C{^1^H} NMR (CDCl_3_): δ 170.6, 169.8, 169.8, 169.7 (4×CO), 132.6 (C-2′), 118.8 (C-3′), 80.9 (C-1), 71.2 (C-4), 69.8 (C-3) 69.0 (C-5), 65.9 (C-2), 62.7 (C-6), 32.8 (C-1′). ESI+-HRMS: [M + Na]^+^ calcd for C_17_H_24_NaO_9_S: 427.1039. Found 427.0994. Physical data matched those published using a related procedure [34].

*(2E)-3-(1,1′-biphenyl-2-propen-1-yl) 2,3,4,6-tetra-O-acetyl-1-thio-**α-D-mannopyranoside (***11***).* To a solution of **9** (53 mg, 0.131 mmol) in dry DCM (2 mL) were added to 4-vinyl-1,1′-biphenyl (**10**) (28 mg, 1.2 equiv), and 10% tricyclohexylphosphine[1,3-bis(2,4,6-trimethylphenyl)imidazol-2-ylidene][2-thienylmethylene]ruthenium(II) dichloride (Grubbs’ catalyst) under N_2_. The solution was refluxed overnight. The solution was evaporated under reduced pressure. The organic layer was separated, dried over Na_2_SO4, filtered, concentrated and the crude residue purified by silica gel column chromatography using Toluene/EtOAc (9.6:0.4) to afford compound 11 as a colorless oil with Rf = 0.24 (Toluene/EtOAc (9.6:0.4)) (41 mg, 0.073 mmol, 56%). [α]^20^D = +70 (c = 0.1, CHCl_3_). ^1^H NMR (300 MHz, CDCl_3_) δ: 7.62 (dd, 4H, ^3^*J*_H-H_ =14.8 Hz, ^4^*J*_H-H_ = 6.3 Hz, H-arom), 7.46 (dd, 4H, ^3^*J*_H-H_ = 15.5 Hz, ^4^*J*_H-H_ = 8.3 Hz, H-arom), 7.38 (dd, 1H, ^3^*J*_H,H_ = 8.6, ^4^*J*_H-H_ = 5.8 Hz, H-arom), 6.56 (d, 1H, *J*_2′,3′_ = 15.6 Hz, CH=CH-biPh), 6.27–6.15 (m, 1 H, C*H*=CH-biPh), 5.4–5.2 (m, 4H, H-3, H-4, H2, H1), 4.41 (m, 2H, H-5, H-6a), 4.15 (dd, 1H, *J*_6a, 6b_ = 12.2 Hz, *J*_5, 6b_ = 2.4 Hz, H-6b), 3.58–3.29 (m, 2H, H-1′a, H-1′b), 2.16, 2.15, 2.08, 2.01 (4s, 12H, 4×COCH_3_). ^13^C{^1^H} NMR (CDCl_3_): δ 170.6, 169.8, 169.8, 169.7, (4×CO), 1405, 135.3, 133.4 (Carom-q), 133.4(C-3′), 129.0 (Carom), 128.2, 127.2, 127.0, 126.9 (Carom), 123.7 (C-2′), 80.7 (C-1), 70.8 (C-4), 69.7 (C-3), 68.9 (C-5), 66.3 (C-2), 62.9 (C-6), 32.6 (C-1′), 20.9, 20.8, 20.7, 20.6 (4×Ac). ESI+-HRMS: [M + Na]^+^ calcd for C_29_H_32_NaO_9_S: 579.1659. Found, 579.1669.

*(2E)-3-(1,1’-biphenyl-2-propen-1-yl) 1-thio-α-D-mannopyranoside (***12***).* The acetylated mannoside **11** was dissolved in dry MeOH (3 mL), a solution of sodium methoxide (1M in MeOH, 0.5 equiv) was added and the reaction mixture was stirred at room temperature until disappearance of the starting material after 5 hr. The solution was neutralized by addition of H^+^ ion-exchange resin (Amberlite IR 120), filtered, washed with MeOH and the solvent was removed in vacuum. The residue was lyophilized to yield the fully deprotected mannoside **12** (20 mg, 0.07 mmol, 73%) after purification by RP-HPLC (see Appendix A). [α]^20^_D_ = +133 (c = 0.2, MeOH). ^1^H NMR (300 MHz, CDCl_3_): δ 7.51 (d, 2H, ^3^*J*_H-H_ = 7.5 Hz,, H-arom), 7.48 (d, 2H, ^3^*J*_H-H_ = 8.1Hz, H-arom), 7.39 (d, 2H, ^3^*J*_H-H_ = 8.1, H-arom), 7.32 (t, 2H, ^3^*J*_H-H_ = 7.6, H-arom), 7.22 (t, 1H, ^3^*J*_H-H_ = 7.3, H-arom), 6.50 (d, 1H, *J*_2′,3′_ = 15.7 Hz, CH=C*H*-biPh), 6.26–6.17 (m, 1 H, C*H*=CH-biPh), 5.14 (s, 1H, H-1), 3.88–3.84 (m, 1H, H-5), 3.82–3.76 (m, 2H, H-4, H-2), 3.66 (dd, 1H, *J*_2, 3_ = 6.0 Hz, *J*_3, 4_= 11.9 Hz, H-3), 3.60–3.54 (m, 2H, H-6a,H-6b), 3.41 (dd, *J*_1′b, 1′a_ = 13.5 Hz, *J*_1′a,2′_= 9.2 Hz, H-1′a), 3.24 (dd, *J*_1′a, 1′b_ = 13.5 Hz, *J*_1′b,2′_= 5.6 Hz, H-1′b). ^13^C{^1^H} NMR (CDCl_3_): δ 141.8, 141.4, 137.1 (Carom-q), 133.3 (C-3′), 129.7 (Carom), 128.2, 127.9, 127.7, 127.6 (Carom), 126.1 (C-2′), 84.6 (C-1), 74.9 (C-5), 73.4 (C-2), 73.3 (C-4), 68.8 (C-3), 62.5 (C-6), 33.1 (C-1′). ESI+-HRMS: [M + NH_4_]^+^ calcd for C_21_H_28_NO_5_S: 406.1688. Found 406.1699.

**X-Ray crystal-structure of 5** and refinement data: formula, (C_21_H_24_O_6_), orthorhombic, space group P212121, a 6.2463 (3) Å, b 7.5145 (3) Å, c 39.1736 (18) Å, *α* 90°, β 90°, γ 90°, V 1838.72(14) Å3, Dcalcd 1.345 g/cm3. Crystallographic data for the structure reported in this paper has been deposited at the Cambridge Crystallographic Data Centre (CCDC) with deposition no: 1840503 for C_21_H_24_O_6_. Appendix A can be obtained free of charge from CCDC, 12 Union Road, Cambridge CB2 1EZ, UK (fax: (+44)1223-336-033; e-mail: deposit@ccdc.cam.ac.uk.

**X-Ray crystal-structure of 6** and refinement data: formula, (C_21_H_24_O_5_), orthorhombic, space group P1 21 1, a 9.7448 (4) Å, b 8.1887 (4) Å, c 21.6060 (9) Å, *α* 90°, β 92.097°, γ 90°, V 1722.94 (13) Å3, Dcalcd 1.374 g/cm3. Crystallographic data for the structure reported in this paper has been deposited at the Cambridge Crystallographic Data Centre (CCDC) with deposition no: 1871374 for C_21_H_24_O_5_. Appendix A can be obtained free of charge from CCDC, 12 Union Road, Cambridge CB2 1EZ, UK (fax: (+44)1223-336-033; e-mail: deposit@ccdc.cam.ac.uk.

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
