# Peer review of "Comparative Study of Aryl O-, C-, and S-Mannopyranosides as Potential Adhesion Inhibitors toward Uropathogenic E. coli FimH"

_molecules, 2019, doi:10.3390/molecules24193566_

Round 1
Reviewer 1 Report
The authors prepared a set of mannopyranosides with 1,1'-biphenyl aglycone linked to the sugar part through O, C and S linkages and studied them as inhibitors for E. coli FimH.
The paper is well written and the data is well presented. I recommend publishing the paper after addressing the following comment:
In absence of a crystal structure for 12, the authors needs to emphasize more on how they confirmed the stereochemistry of compound 12 (or its protected form 11). It is also important to discuss whether the synthesis of 11 using Grubbs catalyst produced exclusively the required isomer or a mixture of E and Z isomers.
Reviewer 2 Report
Authors studied in three mannopyranosides possessing identical to 1,1’-biphenyl glycosidic pharmacophore with three different aglyconic atoms were synthesized and solved X-ray structures, and showed binding abilities against E. coli FimH. Prior to publication, the addition of some experimental data will be essential to strengthen the conclusions of the manuscript.
Authors should show the SPR profiles to check the quality of the data in Fig. 5a and Table 1. Competitive binding studies using SPR is not the novel method, authors should remove the novel word. Authors mentioned “equilibrium could not be reached during injection, KD was obtained from the extracted kinetic rate constants” they used the concentration ranges from 1024 to 8 nM and reported kinetic Kds are about 2.3 and 4.21 nM, is there any reason for not observing steady-state Kd. These compounds are immobilized on CM5 chip, in this method some of the functional groups may not accessible for binding. Did authors try to immobilize FimH1 on SPR chip? Authors should comment on the freedom of the anomeric linkage in three compounds in free and bound states.Author Response
see attachment

Reviewer 3 Report
Major concerns with the manuscript:
The aryl O-, C-, and S-mannopyranosides compared here are described as “possessing identical 1,1’-biphenyl glycosidic pharmacophore [sic]” (lines 17-18 and repeated throughout). This is in fact not the case, as C-glycoside 6 is actually a homologue of the other two, containing one less methylene group. Given that this would place the biphenyl moiety at a different distance from the mannoside core (and even more so considering that C-C bonds are shorter than both C-O and C-S bonds), it is confusing why the authors would consider these identical and attribute the higher affinity of 6 to the C atom without considering/discussing the effects that the removal of a methylene unit would also have. The SPR method used in the study is described as ‘new’ and ‘novel’ (line 202 and throughout). Given that it is a modification of a previous method with the only difference being a change from an octyl to hexyl linker used to conjugate the ligand to the chip surface, it is very similar to previously reported methods. Indeed, it is advantageous (as the authors report) that the reduced hydrophobicity of the hexyl linker facilitates easier regeneration of the chip surface, but it is an incremental improvement to a previous method and should not be described as ‘novel.’ Unfortunately, since the authors have not included the SPR sensorgrams in the Supporting Information it is impossible to evaluate this reported improvement to the regeneration step. The authors have directly compared ligand conformations of X-ray structures of the unbound ligands (crystallized in DCM and/or MeOH) with previously published X-ray structures of ligand bound to FimH. Given the very different effects of crystal packing in protein-free solution, in combination with the fact that crystallization was performed in a more hydrophobic environment, it is a big stretch to infer that the DCM/MeOH-based ligand structures would be representative of the ligand conformation in the FimH binding site (even considering the more hydrophobic nature of the deep FimH binding pocket). The authors also use this crystal structure of 6 (together with computations) to “unambiguously” (line 297) confirm that the C-glycoside exists in its 4C1 chair conformation, which is not supported by the above experiments given how different the conditions used are from the physiological environment.Additional comments:
Lines 57-59: It is unclear what is meant by this sentence, since “several families of potent a-D-mannopyranoside-based antiadhesins” can also be considered to be “small molecule antagonists”. Perhaps the authors are intending to refer to complex oligomannose-based structures, in which case that should be specified here. Lines 71-72: It is unclear what is meant by ‘properties’ here, as “proper pharmacokinetic and pharmacodynamic parameters” are included separately afterwards. Which ‘properties’ are being referred to? Line 74: The phrase “certain with low” should be corrected here. Line 116, Scheme 1: Yields are missing from several steps of the scheme. Lines 186-188: In the first part of the sentence, the Phi angle of compound 12 is reported to be -45.93, but in the last part of the sentence is reported as -46.83. It seems that reference numbers and compound numbers may have been mixed up here, and should be corrected (or further clarified if this is not the case). Line 212, Figure 5: SPR sensorgrams need to be included in the Supporting Information; as is, it is impossible to evaluate quality/reliability of the data. Line 254-256: Presumably after carbodiimide activation the chip surface was passivated with a small molecule (including the reference cell). Perhaps ethanolamine was used as it is the standard, but these additional details should be provided here. Lines 261-265: SPR data/sensorgrams should be included here, especially given the fact that the authors mention not reaching equilibrium. Without this information it is impossible to judge validity of the reported values. Experimental procedures: Mobile phases used must be included with the reported Rf values, or else the Rf values provide no useful information. Specific rotations: Values reported are extremely high for the concentrations given. It should be confirmed that the concentrations reported in the manuscript are in fact reported as c = g/100mL and are not off by a factor of 10. Line 377: Abbreviations have been left out. Supporting Information: Compound numbers should be included in the experimental procedures. Page S4, Line 1: Compound number of the mannoside should be corrected – both 1 and 11 would be incorrect, so its unsure what was meant here. Page S4, Section B: Presumably reverse-phase column chromatography was used here and should be further clarified. Page S4, Section C: “Acetylated” is duplicated. Page S8, NMR for 6-aminohexyl a-D-mannopyranoside: Anomeric proton is reported as a singlet, but 1H NMR spectrum shows two peaks. Page S9, Section 2: Experimental procedure describes the “compound deprotected according to general procedure A” but the compound is in fact acetylated (i.e. not deprotected). Page S12, Section 4. Same comment as above regarding incorrect procedure described (compound is benzoylated).Author Response
see attachment
